# A cultivated planet in 2010: 1. the global synergy cropland map

Miao Lu[1], Wenbin Wu[1], Liangzhi You[1,2], Linda See[3], Steffen Fritz[3], Qiangyi Yu[1], Yanbing Wei[1], Di Chen[4], Peng Yang[1], Bing Xue[5]

[1]Key Laboratory of Agricultural Remote Sensing (AGRIRS), Ministry of Agriculture and Rural Affairs / Institute of Agricultural Resources and Regional Planning, Chinese Academy of Agricultural Sciences, Beijing, 100081, China

[2]International Food Policy Research Institute (IFPRI), Washington, DC, 20005-3915, USA

[3]International Institute for Applied Systems Analysis, ESM, Laxenburg, A-2361, Austria

[4]Institute of Environment and Sustainable Development in Agriculture, Chinese Academy of Agricultural Sciences, Beijing, 100081, China

[5]School of Engineering and Computer Science, Victoria University of Wellington, Wellington, 6140, New Zealand

*Correspondence to*: Wenbin Wu (wuwenbin@caas.cn)

**Abstract**. Information on global cropland distribution and agricultural production is critical for the world's agricultural monitoring and food security. We present datasets of cropland extent and agricultural production in the two-paper series of a cultivated planet in 2010. In the first part, we propose a new Self-adapting Statistics Allocation Model (SASAM) to develop the global map of cropland distribution. SASAM is based on the fusion of multiple existing cropland maps and multilevel statistics of the cropland area, which is independent of training samples. First, cropland area statistics are used to rank the input cropland maps, and then a scoring table is built to indicate the agreement among the input datasets. Secondly, statistics are allocated adaptively to the pixels with higher agreement scores, until the cumulative cropland area is close to the statistics. The multi-level allocation results are then integrated to obtain the extent of cropland. We applied SASAM to produce a global cropland synergy map with a 500 m spatial resolution circa 2010. The accuracy assessments show that the synergy map has higher accuracy than the input datasets, and better consistency with the cropland statistics. The synergy cropland map is available via an open-data repository (DOI: https://doi.org/10.7910/DVN/ZWSFAA. Lu et al., 2020). This new cropland map has been used as an essential input to the Spatial Production Allocation Model (SPAM) for producing the global dataset of agricultural production circa 2010, which is described in the second part of the two-paper series.

## 1 Introduction

Agricultural land satisfies global demands for human food, stock feed, and biofuel, which are increasing at an unprecedented rate with the continuing population and consumption growth (Gibbs et al., 2010; Godfray et al., 2010). Feeding the growing population and meeting this rising consumption remain a great challenge (Kastner et al., 2012; Zhang

et al., 2016; Gao and Bryan., 2017). Accurate spatial information about cropland is vital baseline information for agricultural monitoring and food security (Eitelberg et al., 2015; Yu et al., 2019). Satellite-derived land cover datasets have been widely used for this purpose. For example, the Famine Early Warning Systems Network funded by the United States Agency for International Development has been using cropland distribution and other remote sensing data to provide timely and dependable early warning and vulnerability information related to emerging and evolving food security issues (Brown

and Brickley, 2012). However, there is significant disagreement and high uncertainty among the various land cover datasets (Fritz et al., 2013; Tsendbazar et al., 2015). The uncertainty and inconsistency are particularly high for cultivated lands (cropland and managed pasture) compared to other natural vegetation types, such as tree cover (Congalton et al., 2014). One of the challenges when working with existing cropland datasets is the lack of consistent and reliable data on the location and areal extent of cropland.


Uncertainties and inconsistencies in cropland information are ubiquitous because of the differences in application purposes, cropland definitions and classification methods (Fritz et al., 2013; Verburg et al., 2011; Yang et al., 2017). Globally, spatial agreement in the four global land cover datasets, i.e., IGBP DISCover, the University of Maryland land cover product, the MODIS land cover product, and Global Land Cover 2000 (GLC2000) is about 71.5% (Herold et al., 2008). At the regional

scales, Pérez-Hoyos et al. (2017) compared nine cropland products, including FAO-GLCshare (Food and Agriculture Organization of the United Nations' Global Land Cover Network), GLC2000, GlobCover, Globeland30, and so on, and found that the areas of full agreement in Africa, America, and Asia were only 2.15%, 1.39%, and 11.90%, respectively. Cropland uncertainty is generally higher than that of other land cover classes, especially in transition zones and areas with high landscape fragmentation. For example, disagreements in the Sahelian belt of Africa are prominent because crops are

more scattered and often coexist with grassland (Pérez-Hoyos et al., 2017). In China, the uncertainties and inconsistencies in northwestern and southwestern regions, characterized by high elevations and fragmented landscapes, are higher than those in northern and northeastern areas with more homogeneous landscapes (Lu et al., 2016).

Cropland areas estimated from satellite-based datasets are often inconsistent with statistics, which limits their applications

in agricultural economics and food policy. First, the existing datasets usually focus on the land cover rather than land use because of the direct nature of remote sensing observation (Kerr and Cihlar, 2003; Zeng et al., 2018). Cropland, as an integration of land cover and land use, not only is defined as the crops covering the land surface, but also is influenced by the human activities for food production. However, satellite-based cropland maps may fail to detect cropland features of

land use (Zeng et al., 2018). For example, according to estimates using GlobeLand30, the cropland area in Europe increased

by 22,090 km$^2$ from 2000 to 2010 (Xiang et al., 2018). Yet, the official statistics from FAO indicate a decrease of cropland

in Europe over the same period. One of the main reasons is agricultural land abandonment, which cannot be easily captured

by remote sensing. Secondly, inconsistent definitions of cropland lead to discrepancies between satellite-based estimates

and official statistics. For example, GlobCover 2005/2009, Climate Change Initiative Land Cover (CCI-LC) and MODIS

Collection 5 (MODSI C5) include mosaic classes that mix cropland with other land cover types. Therefore, these products

often under- or overestimate cropland areas, depending on how these mosaic classes are counted (Zeng et al., 2018).

Agricultural statistics are usually collected by interviews and sample surveys, and then computed by aggregating then with

administrative data (Gallego et al., 2010). These statistics provide highly suitable land use information that is not collected

by remote sensing, but often lack spatial details because they are aggregated to the level of administrative units.

Data synergy approaches can take advantage of complementarities between land cover datasets and statistics to solve the

above issues. These approaches can integrate all available satellite-based maps and statistics into a single product, giving

improved accuracy. Synergy approaches are broadly categorized into two types: agreement scoring methods and regression

methods (Lu et al., 2017). The former assumes that the statistical data provide the "true" areas of agricultural land and

spatially disaggregate statistics to pixels according to the agreements of satellite-based datasets. For example, Ramankutty

et al. (2008) used this method to develop global cropland and pasture extent maps of a 1 km spatial resolution circa 2000.

Fritz et al. (2011, 2015) ranked the input datasets and assigned different weights based on their assessed accuracies to

produce the International Institute for Applied Systems Analysis (IIASA)-International Food Policy Research Institute

(IFPRI) cropland map 2005. Regression methods, such as logistic regression and geographically weighted regression

(GWR), establish a regression relationship of cropland percentage between training sample points and input datasets, and

then predict cropland percentage in regions without samples (Brunsdon et al., 1998; Chen et al., 2019). GWR allows

regression parameters to vary over space and has a better fit with the observational data (Chen et al., 2019). GWR has been

used to create global land cover maps and forest maps by using crowdsourced validation data from Geo-Wiki (See et al.

2015; Schepaschenko et al., 2015). However, the above methods generally need sufficient *in-situ* samples for training.

Agreement scoring methods require training samples to assess the qualities of input datasets, and regression models need

training samples to estimate the model parameters at each location. Although crowdsourcing platforms are available for

the sample collection, e.g., Geo-Wiki (www.geo-wiki.org), LACO-Wiki (https://laco-wiki.net), and Collect Earth

(http://www.openforis.org/tools/collect-earth.html), the quality and consistency of samples cannot be assured because the

domain knowledge of the contributors are varied (Bey et al., 2016; Fritz et al., 2009; See et al., 2015).

The objective of this research is to address the issue of training samples for global cropland mapping, and to improve the consistency with statistics and the accuracy of cropland map. We propose a Self-adapting Statistics Allocation Model (SASAM) by fusing multiple statistics and satellite-based cropland datasets to produce a global synergy cropland map. This method is based on agreement among the input cropland datasets, and it is independent of training samples. Cropland area statistics are used to rank the input cropland maps and build a scoring table to indicate the agreement of the input

datasets. Statistics at the national, first and second subnational levels are allocated to the pixels with higher cropland scores, and then the results are integrated to obtain the cropland extent. Using this method, we have produced a global cropland synergy map circa 2010 with a spatial resolution of 500 m. The remainder of this paper is organized as follows. We present the input data sources in Section 2 and describe the SASAM in detail in Section 3. The results and analysis are presented in Section 4, data accessibility is described in Section 5, followed by the discussion and conclusion in Section 6.

**2 Data sources**

The data sources used in this study include global and regional satellite-based cropland products and multilevel statistics for cropland areas.

**2.1   Satellite-based maps and data pre-processing**

At the global scale, five cropland products around 2010 were selected from GlobeLand30, CCI-LC, GlobCover 2009,

MODIS C5, and the Unified Cropland Layer (Table 1). GlobeLand30 was produced from Landsat images and China HJ images by using the Pixel-Object-Pixel (POK) classification method (Chen et al., 2015). CCI-LC and GlobCover 2009 were generated by the European Space Agency (ESA) with similar classification strategies of unsupervised clustering and supervised learning (Bontemps et al., 2017; Defourny et al., 2017). MODIS C5 was generated from MODIS time series data using the decision tree method (Friedl et al., 2010). The Unified Cropland Layer is a hybrid map based on a

combination of the fittest products according to four dimensions: timeliness, legend, resolution, and confidence (Waldner et al., 2015).

At the regional scale, we selected publicly available products with high spatial resolution and quality in Europe and North America (Table 1). CORINE Land Cover (CLC) 2012 covers 39 European countries with a total area of over 5.8 million

$km^2$. CLC2012 is an update of CLC2006 developed using computer-assisted photointerpretation of high-resolution satellite

images from 2011 and 2012 (Hościło & Tomaszewska, 2015). The North American Land Change Monitoring System, cooperating with Natural Resources Canada, the United States Geological Survey, and three Mexican organizations, produced the 2010 North American Land Cover 30 m dataset for Canada, USA, and Mexico. Each country developed its own classification method to identify land cover classes and then provided an input layer to produce a continental land cover map across North America.


In addition, we collected land cover maps in two countries, i.e., Australia and China, as supplements. The Land Use of Australia 2010–2011 was produced by the Australian Bureau of Agricultural and Resource Economics and Sciences operated under the Australian Government Department of Agriculture, and the agricultural land use data are based on the Australian Bureau of Statistics' 2010–2011 agricultural census data (Smart, 2016). The National Land Use/cover Database of China (NLUD-C) 2010 was updated from NLUD-C 2008 based on images with approximately 30 m spatial resolution using visual interpretation, field surveys, and large amounts of auxiliary information (Zhang et al., 2014).


Pre-processing of these satellite-based maps was essential because of their differences in coordinate systems, spatial resolution, and classification schemes. First, we masked non-agricultural areas in the satellite datasets. Then, the geographic latitude/longitude coordinate system with WGS84 datum was chosen as the base projection for coordinate transformation. Because the spatial resolutions of regional and global products vary from 30 m to 500 m, a standard geographical grid with 0.0041667° (i.e., about 500 m) resolution was employed to aggregate the input products with cropland percentages.


(insert Table 1 here)
Table 1: Input satellite-based products.

The critical part of the data pre-processing is the cropland definition harmonization. We used FAO's definition of cropland as "arable lands and permanent crops." Arable land is the land under temporary agricultural crops (multiple-cropped areas are counted only once), temporary meadows for mowing or pasture, land under market and kitchen gardens, and land temporarily fallow (less than five years). Permanent crops are the land cultivated with long-term crops which do not have to be replanted for several years (such as cocoa and coffee), land under trees and shrubs producing flowers (such as roses and jasmine), and nurseries (except those for forest trees, which should be classified as "forest"). Abandoned land resulting from shifting cultivation and permanent meadows or pastures are excluded from cropland in our study. The cropland-related


classes of each dataset were extracted given percentage weights according to their cropland definition: pure cropland classes were assigned higher percentage weights, and mosaic cropland classes were assigned lower weights (Lu et al. 2017). Through this process, we produced cropland percentage maps derived from each satellite-based product at a 500 m resolution with the same coordinate system.

## 2.2 Statistics of the cropland area

We collected statistics of the cropland area at the national, first and second subnational levels circa 2010. The national statistics were acquired from FAO's FAOSTAT Land Use database (http://www.fao.org/faostat/en/#data/RL), which covers about 200 countries and territories of the world. The statistics are widely useful for market management, production forecasts, and policy-making in the agricultural and food sectors. Following our adopted cropland definition, the item "Arable lands and permanent crops" was selected from the statistics. Because the satellite-based products were mainly from 2009 to 2011, the average values from 2009 to 2011 were calculated to provide more stable estimates for the synergy cropland in 2010. The cropland area statistics available at the national level are shown in Fig. 1(a), which covers almost all countries in the world.

While statistics of the national cropland area are available from FAO, subnational statistics are not provided by a single multinational institution, and they are rarely available at the global scale. Nevertheless, for several decades, IFPRI and its partners have collected the subnational agricultural statistics on cropland and individual crops in many countries throughout the world, and paid particular attention to developing countries in Africa, Latin America, and Asia. If a cropland value exists for a subnational unit, this value is taken and the harvested areas of individual crops within the unit are ignored. Otherwise, the cropland area is calculated by adding the harvested areas of all crops growing within the administrative unit divided by the cropping intensities of the individual crops. The cropping intensity varies by rainfed or irrigated systems and by countries. The intensity data were collected from various sources such as seasonable harvested area, expert judgments and household surveys (Yu et al., 2020). Because of possible missing areas or missing crops, the cropland value at the subnational level is a minimum estimate of the actual cropland of that unit.

(insert Fig. 1 here)

Figure 1: The statistics of cropland area at the national (a), first subnational (b), and second subnational (c) levels.

There are two levels of subnational statistics. The first subnational level indicates a lower unit than the national administrative division, such as provinces in China or Canada, and states in the United States or India. We collected the statistics for 64.91% of the first subnational units in most countries, not in a few countries in Africa (Fig. 1(b)). The second subnational level indicates smaller administrative units such as prefecture-level cities of China, counties of the United States, and departments of France. Statistics for 34.76% of the second subnational units were obtained (Fig. 1(c)).

## 3 Methodology

The principle of SASAM is to automatically allocate the cropland area taken from the statistics to the pixels with higher cropland likelihood. The cropland distribution is adjusted adaptively until the cumulative cropland area is close to the statistics. The model has three main steps, i.e., agreement ranking establishment, self-adapting statistics allocation, and integration of multilevel allocation results. First, the national statistics are used to assess the accuracies and set weights for the satellite-based cropland input maps, and then a scoring table is built based on the weights of the input maps to generate agreement ranking results. The national and subnational statistics are self-adaptively allocated to the pixels according to their agreement ranking. Lastly, the allocated results are integrated to generate a synergy cropland map.

### 3.1 Agreement ranking establishment

Generally, the higher agreement among input datasets indicates a higher likelihood of cropland. The assessed accuracies of the input datasets also affect synergetic confidence (Fritz et al., 2015; Lu et al., 2017). We use the national statistics to assess the accuracies of satellite-based datasets, and then adaptively establish agreement ranking scores according to the accuracies and agreements of the input datasets.

For each input dataset, the cropland area in each country is estimated as:

$$a_{i,j} = \sum_{n=1}^{N}(m_n \times p_n) \tag{1}$$

where $a_{i,j}$ is the cropland area of country $j$ estimated by input dataset $i$, $n$ is the pixel labeled as cropland, and $P_n$ is the percentage of cropland in pixel $n$ after data processing. Because we use a geographic latitude/longitude coordinate system, the pixel area $m_n$ is calculated by equal-area projection (Lu et al., 2017). Then the absolute difference $Diff_{i,j}$ between the cropland area estimated from input dataset $i$ and the statistics is calculated to assess the accuracy of the input map, as shown in Eq. (2):

$$Diff_{i,j} = abs\left(\frac{a_{FAO,j} - a_{i,j}}{a_{FAO,j}}\right) \tag{2}$$

where $a_{FAO,j}$ is the cropland area statistics of country $j$ derived from FAO. A lower value of $Diff_{i,j}$ indicates better agreement with the official statistics, and a higher ranking for the input map.

An agreement ranking score is established using a table reflecting the agreement and rankings of the input datasets. If there are five input datasets, their rank from the highest to the lowest are labeled A, B, C, D, and E (Table 2). The agreement levels ranging from 0 to 5 indicate the number of input datasets identifying a pixel as cropland. Because there are 32 permutations for the five input datasets ($2^5 = 32$), the scores are from 0 to 31. A higher score value indicates a higher likelihood of cropland. The agreement level of 5 means that all the input datasets identify the pixel as cropland and the pixel has the highest score of 31, while the agreement level 0 indicates that all the datasets classify the pixel as non-cropland and the pixel has the lowest score of 0. For other agreement levels, there are various permutations. For example, when the agreement level is 4, there are five combinations for the datasets with score values set from 26 to 30. Because A, B, C, and D have higher rankings, if all four indicate cropland, then the score value is set as 30, which is higher than other combinations. According to these rules, we obtained values for the full scoring table with five input datasets (Table 2). Similarly, we utilized this method to obtain the scoring table ranging from 0 to 63 with six input datasets. The scoring table is then used to transform the input cropland layers into an agreement ranking map. Meanwhile, the average cropland percentages of the input datasets are calculated with a spatial resolution of 500 m.

(insert Table 2 here)

Table 2: The ranking scoring table for five input datasets.

**3.2 Self-adapting statistics allocation**

The self-adapting statistics allocation is to allocate cropland area statistics to the pixels with higher ranking scores automatically, and this process is adjusted adaptively until the cumulative cropland area is close to the statistics. Figure 2 shows the flowchart of statistics allocation with five input datasets as an example. First, the pixels with the highest score of 31 are selected, and their total area is calculated by Eq. (3):

$$A_{31} = \sum(m_{31,n} \times p_{31,n}) \tag{3}$$

where $m_{31,n}$ and $p_{31,n}$ are the pixel area and average percentage of pixel $n$ labeled as the score 31. Then the area is

compared with the statistics. If the area is much smaller than the statistics, the cropland pixels with the next lower agreement ranking, such as 30, are chosen, and the total area is then calculated as in Eq. (3). The cumulative cropland area with the score of 30 and above is compared with the statistics. If the cumulative area is very close to the statistics, the pixels labeled

with scores of 31 and 30 are selected as cropland pixels. Otherwise, pixels with lower scores are selected and added until the cumulative area reaches the statistics. In Fig. 2, when the cumulative area with the score 29 is the closest to the statistics, the pixels with score values from 29 to 31 are selected as the cropland extent. We obtain the cropland percentages and scores of the cropland pixels. The values of the scores indicate the agreements of the input cropland datasets, which reflects the confidence level of the cropland pixel. The scores range from 0 to 31 for five input datasets, and from 0 to 63 for six

input datasets. Therefore, min-max normalization is used to normalize the scores to the same scale. The normalization results are the confidence levels with values from 0 to 100%.

(insert Fig. 2 here)

Figure 2: The flowchart of cropland area statistics allocation with five input products.


Allocation results include the score values and the average percentage maps comprising the selected cropland pixels. Using the above method, we allocated the national, first and second subnational statistics to the pixels respectively, and obtained multilevel allocation results.

**3.3 Integration of multilevel allocation results**

The qualities of the cropland area statistics are various. At the national level, the FAO statistical system includes a quality framework and a mechanism to ensure the compliance of FAO statistics to this framework. Therefore, it is reasonable to consider that national statistics have higher reliability. Subnational statistics are estimated by the harvested crop areas and the cropping intensity factors when the official statistics are unavailable. In some subnational units, especially at the second

subnational level, only a few harvested areas of some crops are available, so the estimated cropland areas may be much lower than the actual cropland amount (You et al., 2014; Fritz et al., 2015). Meanwhile, some cropland area statistics are absent in subnational units. We collected the statistics for 64.91% of the first subnational units, and 34.76% of the second subnational units (Fig. 1). Therefore, it is reasonable to consider that the national statistics are more reliable than the subnational ones, and the first subnational statistics are more reliable than the second ones. The integration principle is that

the overall cropland area at the national level should be consistent with the statistics, and the cropland area of the lower level should be equal to or greater than the statistics.

We take San Luis Province in Argentina as an example to describe the integration process. The first and second subnational allocation results with cropland are shown in Fig. 3(a) and (b). This province consists of nine departments, labeled A-I in

Fig. 3(b). The cropland areas of the second subnational allocation results in departments C, D, E, F, and G are 0 because of the absence of the second subnational statistics. The cropland areas of the first subnational allocation results in each department are calculated (Table 3). The integration of the first and second subnational allocation results uses the following rules:

(1). For the departments which have statistics, when the cropland area in the second subnational unit is higher than the area at the first subnational level, the second subnational allocation results are used for this department. Otherwise, the first subnational allocation results are used. As shown in Table 3, the total cropland area of the second subnational units (692.09 $km^2$) in the department I is higher than that for the first subnational area (291.46 $km^2$). The result for the second subnational units is selected as the allocation result for department I. For departments A, B, and H, the results of the two levels are the

same, and the allocation is unchanged (Fig. 3(c), Table 3).

(2). Next, the departments with no statistics are merged. The cropland area differences between the first and second subnational allocation results are calculated and allocated to the merged departments. For example, in Fig. 3, the total cropland area of the first subnational allocation results and the second subnational results are 4,909.10 $km^2$ and 4,144.12

$km^2$, and their difference, 764.98 $km^2$, is allocated to the merged departments of C, D, E, F, and G (Fig. 3(c), Table 3).

(3). The self-adapting statistics allocation in Section 3.2 is rerun for the merged departments of C, D, E, F, and G with a cropland area 764.98 $km^2$. Based on the agreement ranking scores established in Section 3.1, the cropland area 764.98 $km^2$ is allocated to the pixels with higher ranking scores automatically until the cumulative cropland area is close to the 764.98

$km^2$. Then, we obtained the allocation results of the merged region, as shown in Fig. 3(d).

According to the above integration rules, we first integrated the first and second subnational results to obtain subnational

cropland results, and then combined the subnational and national allocation results to create the final synergy cropland map.


(insert Fig. 3 here)

Figure 3: The integration of the first and second subnational allocation results in San Luis Province, Argentina: (a) the first subnational allocation result, (b) the second subnational allocation result, (c) the combination of the departments with no statistics, and (d) the allocation results of the departments with no statistics.


(insert Table 3 here)

Table 3: Cropland areas of each department from the first and second subnational allocation results, and their coordination in San Luis Province of Argentina.

**3.4  Validation of the global cropland map and comparison with IIASA-IFPRI method**

The accuracies of the spatial location and cropland area for the global cropland map were assessed. The percentage cropland map was first reclassified into a binary map of cropland/no cropland, where a cropland percentage greater than zero was assigned to the cropland category. The spatial accuracies were assessed by using an error matrix based on training samples. These samples originated from the Tsinghua University in their development of the FROM-GLC land cover product (Gong

et al., 2013). The samples types were identified manually by hundreds of students, researchers, and experts using Google Earth images in or around 2010. We selected the samples between 70°N and 60°S where almost all cropland in the world lies. The test data consisted of 5,743 cropland samples and 28,076 non-cropland samples. The cropland areas of cropland maps were calculated in each country, and then compared with FAO statistics using the correlation coefficient (R) and root mean square error (RMSE) to assess the consistency.


We compared the SASAM with the IIASA-IFPRI method (Fritz et al. 2015) in China. Unlike SASAM, the IIASA-IFPRI method needs training samples to assess the accuracies of input datasets for building the weighted scoring table (Fritz, et al., 2015). Training samples from China (1,387 cropland and 1,430 non-cropland) were employed to assess the accuracies of the input datasets. Then, the spatial location and the cropland area accuracies for the results of SASAM and IIASA-

IFPRI method were calculated and compared.

## 4 Results and analysis

### 4.1 Results of global synergy cropland

Agreement ranking was used to generate scores and average cropland percentages for the satellite-based input data. The ranges of scores were determined by the amount of the input datasets. Regional cropland maps in Europe, USA, Canada, Mexico, Australia, China, and South Africa were available, so agreement ranking scores ranged from 1 to 63. The agreement ranking score map with values from 1 to 63 is shown in Fig. 4(a) for Europe. In the other regions, e.g., Africa (Fig. 4(c)), the scores ranged from 1 to 31 with the five global input datasets used for cropland synergy. Meanwhile, average cropland percentages were obtained by taking the mean percentages of the input datasets. Maps for Europe and Africa are shown in Fig. 4(b) and Fig. 4(d), respectively. The areas with higher scores usually have higher average cropland percentages.

(insert Fig. 4 here)

Figure 4: Agreement ranking score maps and average cropland percentages in Europe and Africa: (a) and (b) are the score map and cropland percentage of Europe; (c) and (d) are the score map and cropland percentage of Africa.

After the agreement rankings were determined, the statistics were allocated to pixels with higher scores, and then the national, first and second subnational statistics allocation results were obtained. In Europe, all the national statistics were collected, and the national synergy results are shown in Fig. 5(a). We obtained the first subnational statistics for 510 out of the 586 administrative units (87.03%), and the second subnational statistics for 951 out of the 3,313 administrative units (28.71%). Therefore, the cropland extent of the national level is greater than that of the subnational level, and the first subnational level has more cropland extent than the second subnational level (Fig. 5(a–c)). In Africa, the national synergy results are shown in Fig. 5(d). At the first subnational level, 618 of the 796 administrative units have statistics (77.63%). We did not have the first subnational statistics for Central African Republic, Congo, Seychelles, Libyan, Equatorial Guinea, Eritrea, Western Sahara, and Cape Verde. Therefore, in these countries, there are no the first subnational synergy results. At the second subnational level, only 13.89% (770 out of the 5541) of the administrative units have statistics. About 37 countries, including Nigeria, Sudan, and Namibia, do not have the second subnational statistics. As a result, the corresponding areas do not have allocation results (Fig. 5(f)).

(insert Fig. 5 here)

Figure 5: Statistics allocation results in Europe and Africa: (a) and (d) are the national allocation results; (b) and (e) are the first subnational allocation results, (c) and (f) are the second subnational allocation results.

The allocation results of the national, first and second subnational levels were integrated using the rules described in Section 3.3. First, the first and second subnational allocation results were combined to obtain the subnational allocation results, and then the results were integrated with the national allocation results to generate the final synergy cropland map at the global scale (Fig. 6(a)). The confidence level map of synergy results was created by normalizing the agreement ranking scores of the synergy cropland pixels (Fig. 6(b)). The results indicate that India, China, America, Russia, Kazakhstan, and Ukraine have large cropland areas. Latin America is becoming an important grain-producing area because new agricultural land has been established from intact and disturbed forests since the 1980s (Gibbs et al., 2010). The higher confidence levels are usually in homogeneous areas, while lower confidence levels are in areas with heterogeneous landscapes or at the margins of cropland extent (Fig. 6(b)).

(insert Fig. 6 here)

Figure 6: The results of global synergy cropland: (a) cropland percentage map, (b) confidence level of synergy cropland.

### 4.2 Accuracy assessments and analysis

### 4.2.1 Spatial accuracy assessment

The spatial accuracies of the five global input datasets and the synergy cropland map were assessed at the continent and global scales (Table 4). The accuracy of the synergy cropland mapping is 90.8%, which is higher than those of the five input datasets at the global scale. In North America, Europe, Oceania, and Asia, the overall accuracies are 92.4%, 93.7%, 96.5%, and 88.3% respectively, which are higher than any of the five input datasets. In South America, the accuracy of the synergy cropland (89.4%) is somewhat lower than GlobeLand30 (90.1%). Also, in Africa, the accuracy of synergy cropland (89.1%) is slightly lower than GlobeLand30 (89.9%). In North America, Europe, Oceania, and Asia, the regional cropland data are available, while the regional datasets are unavailable in South America and Africa. This is one reason why the accuracies of the synergy results in South America and Africa are slightly lower than some of the input datasets.

(insert Table 4 here)

Table 4: Overall accuracies of input datasets and synergy cropland at the continent and global scales.

**4.2.2 Statistical consistency**

The cropland areas of the global input datasets and the synergy cropland map in each country were calculated and correlated with the statistics (Fig. 7). The correlation coefficient of the synergy map is 0.99 and higher than any of the input datasets (Fig. 7(f)). The high correlation is because the synergy map is produced by the fusion of statistics and landcover maps. GlobeLand30 and MODIS Collection 5 have higher correlation coefficients (0.97) than other input datasets, while CCI-LC and GlobCover have lower correlation coefficients (0.88 and 0.89, respectively). In addition, RMSE is used as another indicator to assess the dispersion between the cropland maps and the statistics. Although the correlation coefficients of the synergy cropland map, GlobeLand30, and MODIS C5 are similar, the RMSE of the synergy cropland ($3.41 \times 10^4$) is much lower than that of GlobeLand30 and MODIS C5, which are $8.75 \times 10^4$ and $7.03 \times 10^4$, respectively. Therefore, the synergy map has the best consistency with the national statistics.

(insert Fig. 7 here)

Figure 7: The consistency analysis between cropland areas estimated from products and statistics: (a) GlobeLand30, (b) Unified Cropland, (c) CCI-LC, (d) GlobCover 2009, (e) MODIS C5, and (f) synergy map.

The cropland areas of the synergy map are higher than the statistics in some countries (Fig. 7). SASAM is a process that accumulates cropland areas from high to low scores until the accumulated area reaches the statistics. Because cumulative areas are not continuous, the cropland area estimated by the synergy map might not be very close to the required statistics. Sometimes the difference may be substantial. For example, in Japan's case, the national statistics for the cropland area is 45,977.50 km$^2$. The accumulated cropland areas with scores above 27 and above 26 are 40,618.13 km$^2$ and 52,867.19 km$^2$, respectively. If we take all pixels with scores above 26, the national area estimated by the synergy map (52,867.19 km$^2$) is almost 15% more than the national statistics. Meanwhile, in a few countries, such as Niger, Saudi Arabia, and Dominica, the areas of synergy cropland are slightly lower than the statistics. This is because the cropland areas estimated from the input datasets are all lower than the statistics. For example, in Niger, the cropland area of national statistics is 152,250 km$^2$, while the cropland areas estimated by GlobeLand30, Unified Cropland, CCI-LC, GlobCover, and MODIS C5 (i.e., 66,163 km$^2$, 140,259 km$^2$, 139,734 km$^2$, 21,925 km$^2$, and 76,018 km$^2$, respectively) are all smaller than the statistics. The synergy

map is based on these input cropland layers, and so the synergy cropland area, 140,022 km$^2$, is inevitably smaller than the statistics.

### 4.2.3 Comparison with the IIASA-IFPRI method

For the IIASA-IFPRI method, the rankings from high to low are MODIS C5, Unified Cropland, CCI-LC, NLUC-C, GlobeLand30, and GlobCover by using the training samples in China. The input datasets were ranked according to their accuracies for the scoring table, and then national statistics were allocated to the pixels with higher scores (Fritz et al., 2015). From the highest score of 63, the accumulated area was calculated until the score of 59 where the cropland area was closest to the statistics of $1.23 \times 10^6$ km$^2$ (Table 5(a)). At the same time, SASAM was employed for synergy cropland estimation using the same input datasets and statistics. The cropland areas of the input datasets were estimated and compared with statistics for ranking, giving the ranks from high to low as MODIS C5, NLUC-C, GlobeLand30, Unified Cropland, CCI-LC, and GlobCover. The accumulated area was calculated from the score of 63 to the score of 58, which is closest to the statistics (Table 5(b)).

There are a few slight differences between the results derived from the IIASA-IFPRI and SASAM methods. Validation samples, i.e., 1403 cropland and 1430 non-cropland, were employed to compare the accuracies of the results. The overall accuracy of the IIASA-IFPRI result is 77.68%, and that of SASAM is 77.75%. The cropland areas estimated from the IIASA-IFPRI method and SASAM are $1.23 \times 10^6$ km$^2$ and $1.25 \times 10^6$ km$^2$, respectively, which are both consistent with the national statistics of $1.23 \times 10^6$ km$^2$. The comparison in Table 5(a) and (b) shows that the selected combinations of input datasets are similar, except that SASAM has one more combination with the score of 58. SASAM provides excellent performance without training samples, which is a cost-effective way to map cropland using the synergy between datasets.

(insert Table 5 here)

Table 5: Calculation of accumulated areas from high score value to low: (a) IIASA-IFPRI method, (b) SASAM.

### 5 Data and code accessibility

The global cropland map and the confidence level map are open access and available at: https://doi.org/10.7910/DVN/ZWSFAA (Lu et al., 2020). The subnational statistics of cropland area are available at: https://doi.org/10.7910/DVN/PRFF8V (International Food Policy Research Institute, 2019). All the code with annotations used for the synergy cropland mapping is shared at this website: https://sourceforge.net/projects/globalmapping/ .

## 6 Discussion and conclusion

The cropland areas estimated from satellite-based products are generally inconsistent with statistics, which hinders the application of cropland maps in some studies, such as food security, agricultural sustainability, and the carbon cycle. In this study, a synergy method (SASAM) was developed to produce a new global cropland map for the year 2010 with a 500 m spatial resolution. Our research makes two contributions to cropland mapping at the global scale. First, SASAM addresses the issue of requiring lots of training samples for global cropland mapping. Secondly, we have considerably improved the

accuracy of the final cropland map for 2010, which is consistent with official statistics.

SASAM does not rely on training samples, which is more cost-effective for cropland mapping. Traditional synergy methods usually need a relatively large amount of training samples to assess the accuracy of the input datasets. Although the crowdsourcing tools, such as Geo-Wiki, provide a new low-cost way of gathering samples, quality and uncertainty issues

cannot be ignored because the samples are collected mostly by volunteers. Our method uses official statistics as the reference to assess the accuracies of the input datasets. Datasets with higher accuracies generally have greater consistencies with statistics (Lu et al., 2015, 2016). For example, the accuracies of GlobeLand30 and MODIS C5 are higher, and their consistencies are also better than other input datasets. By contrast, GlobCover has lower overall accuracy and consistency with statistics (Table 4 and Fig. 7). Hence, statistics can replace training samples to assess the input datasets. The

comparison with the IIASA-IFPRI method in China confirms that SASAM, without training samples, performs well in cropland synergy.

The accuracy of the synergy cropland map and its consistency with statistics are higher than the input datasets. At the global scale, the accuracy of the synergy cropland mapping (90.8%) is higher than the five input global datasets. At the regional

scale, the continents with regional input datasets, such as North America, Europe, Oceania, and Asia, have the highest overall accuracies. For the continents without regional datasets, such as South America and Africa, the accuracies of the synergy cropland are a little lower than GlobeLand30. Therefore, the regional datasets are essential for improving the accuracy of the synergy map. The higher correlation coefficient and lower RMSE indicates that the synergy map has better consistency with statistics than the input datasets. SASAM is a process that selects pixels with a high likelihood of cropland

until the cumulative area reaches the statistics. The synergy map combines the advantages of land cover products and statistics, taking into account the land use and land cover characteristics for cropland.

The cropland areas estimated by the synergy map are close, but not exactly equal to the statistics. The scoring table is discrete, and its values range from 0 to $2^n-1$ where n is the number of input datasets. The agreement ranking scores are from 0 to 31 for the five input datasets, and from 0 to 63 for six input datasets. The cumulative cropland area is calculated from high to low scores until it is close to the statistics. The final cumulative area is slightly higher than the statistical areas to further support the spatial production allocation model (SPAM), which is described in the second part for the two-paper series of a cultivated planet in 2010. The allocation rule can be adjusted to suit various applications of cropland mapping. If the synergy result needs to be strictly consistent with the statistics, the closest cumulative area, which may be lower than the statistics, can be selected. We employed the national, first and second subnational statistics for SASAM. Subnational statistics are critical, especially for large countries such as India, China, and the USA, because the subnational statistics not only consider the spatial heterogeneity of cropland distribution, but also reduce the allocation errors from the national statistics.

Although we have shown that cropland extraction from multiple sources in this study is efficient, we also recognize that there are uncertainties associated with this approach. First, the agricultural landscape is an essential factor affecting the agreements of the input datasets for the cropland synergy map. In homogeneous areas, the high agreements among the input datasets are dominant, so the selected cumulative areas have high agreement ranking scores, such as India, America, Argentina, and Brazil. In heterogeneous areas, the agreements of the input datasets are lower, so the synergy results have more uncertainties. Secondly, differences in the cropland definition can also affect the agreement among the input datasets. For CCI-LC and GlobCover, some mosaic classes of cropland and forest are common in hilly areas. For example, in Indonesia, Malaysia and Philippines, CCI-LC and GlobCover classified permanent crops (coffee, cocoa, and rubber) as cropland, while GlobeLand30 classified these as forests. Besides, because pastures have similar features with cropland, GlobeLand30 employing textural and spectral features for classification usually classifies pastures as cropland. Therefore, the cropland synergy map has uncertainties in farming-pastoral zones. Thirdly, subnational statistics at the global scale were collected from multiple sources, and uncertainties are high because of differences in data processing and quality criteria across countries. In Europe, America, Canada, China, and other regions, the official censuses of cropland area at subnational level are available and reliable. While the cropland areas are the ratios between harvested areas of all crops and the cropping intensities, in some developing countries of Africa, Latin America, and Asia, the cropland area statistics in these regions are less reliable because of possible missing harvested areas of crops.

We will collect more reliable input data and explore the integration of synergy approach and machine learning in the future to solve the above uncertainties and further improve the quality of the cropland dataset. The quantity and quality of the input datasets are the basis of the synergy approach. We will collect more existing cropland maps with a high spatial resolution to refine the agreement ranking scores. SASAM accumulates cropland areas from high to low scores until the accumulated area reaches the statistics. The cumulative cropland area will be closer to the statistics with more input cropland datasets. Meanwhile, we will collect more statistics of cropland area at the subnational level. If all the subnational statistics at the global scale were available, the integration of multilevel allocation results would not be needed, which would greatly simplify the synergy process. To improve the method, we will explore the integration of synergy approach and machine learning according to the agreement of the input data and the geographical landscape. The synergy method is economical and efficient for cropland mapping in those regions with highly homogeneous landscapes. The regions with heterogeneous landscapes usually have lower agreements with higher uncertainties. Therefore, we will employ deep learning for cropland classification based on using high spatial resolution images with training samples from the agreements of existing cropland maps.

We applied SASAM to produce a global cropland map for 2010 with a 500 m spatial resolution. The synergy map has higher accuracy and better consistency with statistics than the original datasets, and it combines the advantages of the land cover products and statistical datasets. Therefore, the map can better support relevant studies such as hydrological modeling, land use assessment and agricultural monitoring. In particular, the current synergy cropland dataset underpins the development of SPAM2010: the latest global gridded agricultural production maps in 2010, which is introduced in the second part of the two-paper series (Yu et al., 2020). Although some products of more recent years are available, such as CCI-LC for 2015, the quantity of the input datasets is still not sufficient to support SASAM to produce a more recent cropland map. With the development of new individual cropland maps, we will update the synergy cropland map in the future and further improve the accuracy of synergistic mapping, especially in regions with heterogeneous landscapes.

**Author contributions**

Miao Lu, Liangzhi You, Linda See, and Steffen Fritz designed the experiments. Miao Lu, Wenbin Wu, Qiangyi Yu, and Peng Yang carried them out to develop the cropland map. Miao Lu developed the model code. Yanbing Wei and Di Chen conducted the validation work. Liangzhi You, Wenbin Wu, Linda See, and Bing Xue did writing-reviewing and editing. Miao Lu prepared the manuscript, and wrote the final paper with contributions from all the co-authors.

**Competing interests**

The authors declare that they have no conflict of interest.

**Acknowledgments**

The work is financially supported by the National Key Research and Development Program of China (2019YFA0607400), the National Natural Science Foundation of China (41921001), and the Fundamental Research Funds for Central Non-profit Scientific Institution (1610132020016). This activity forms part of the CGIAR Research Program on Water, Land and Ecosystems led by International Water Management Institute (IWMI) and the CGIAR Research Program on Policies, Institutions, and Markets (PIM) led by International Food Policy Research Institute (IFPRI).

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

## 625  Figures and tables

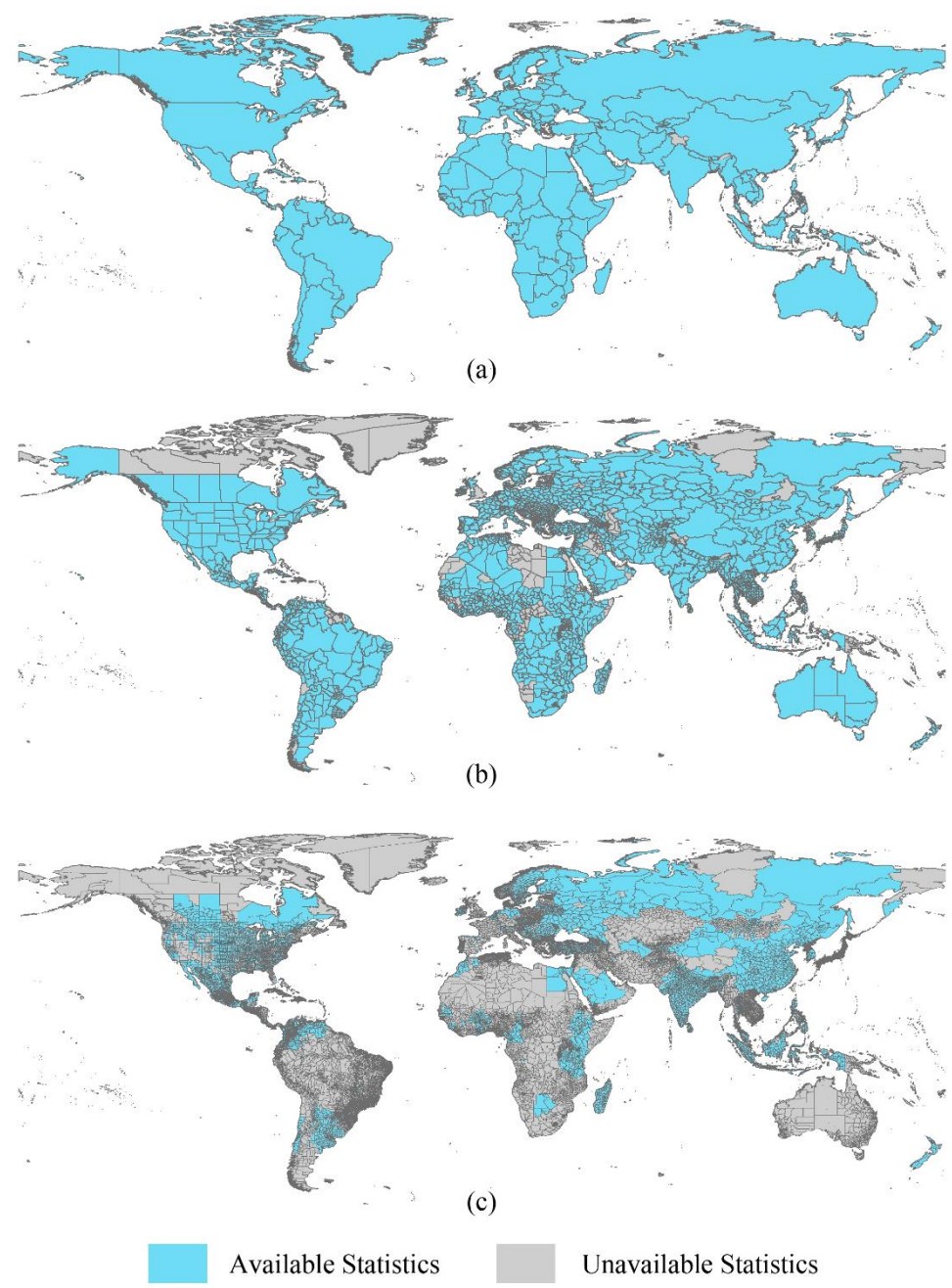

(a)

(b)

(c)

Available Statistics          Unavailable Statistics

**Figure 1: The statistics of cropland area at the national (a), first subnational (b), and second subnational (c) levels.**


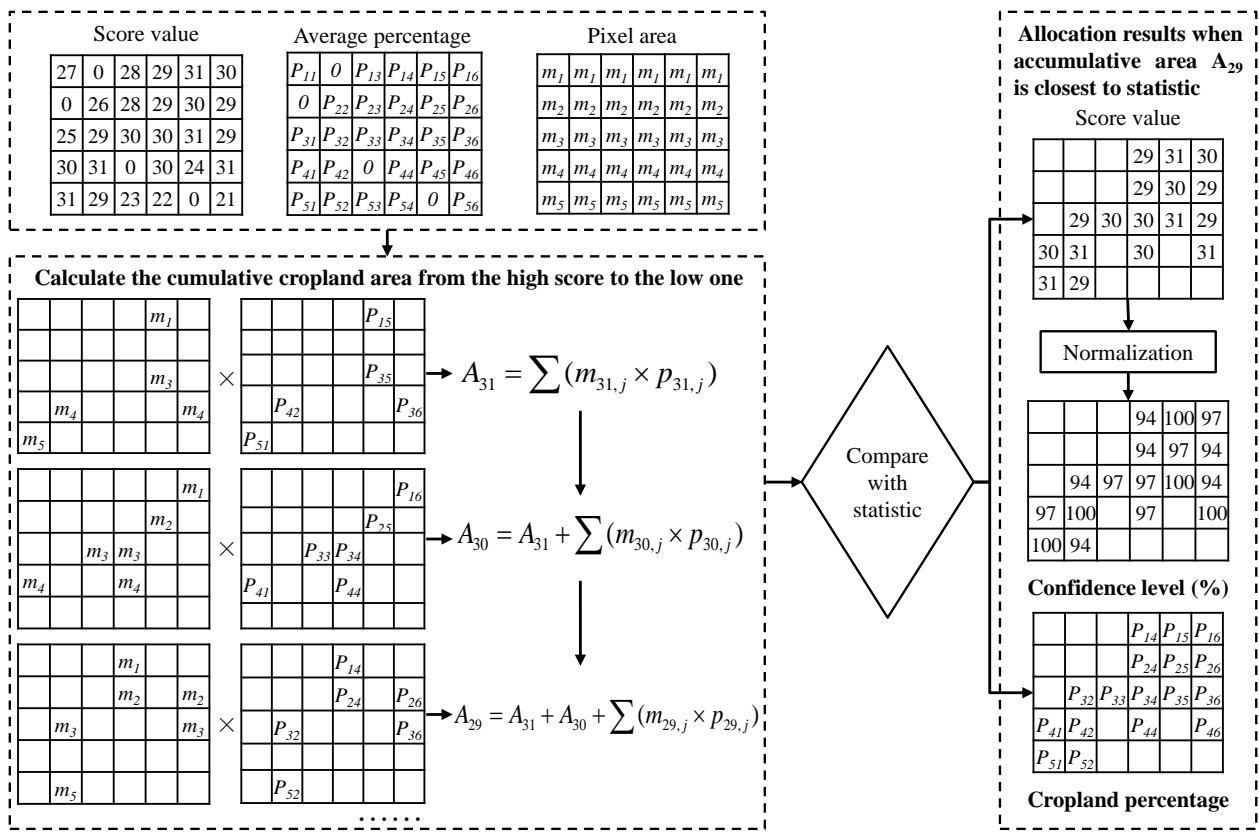

**Figure 2: The flowchart of cropland area statistics allocation with five input products.**


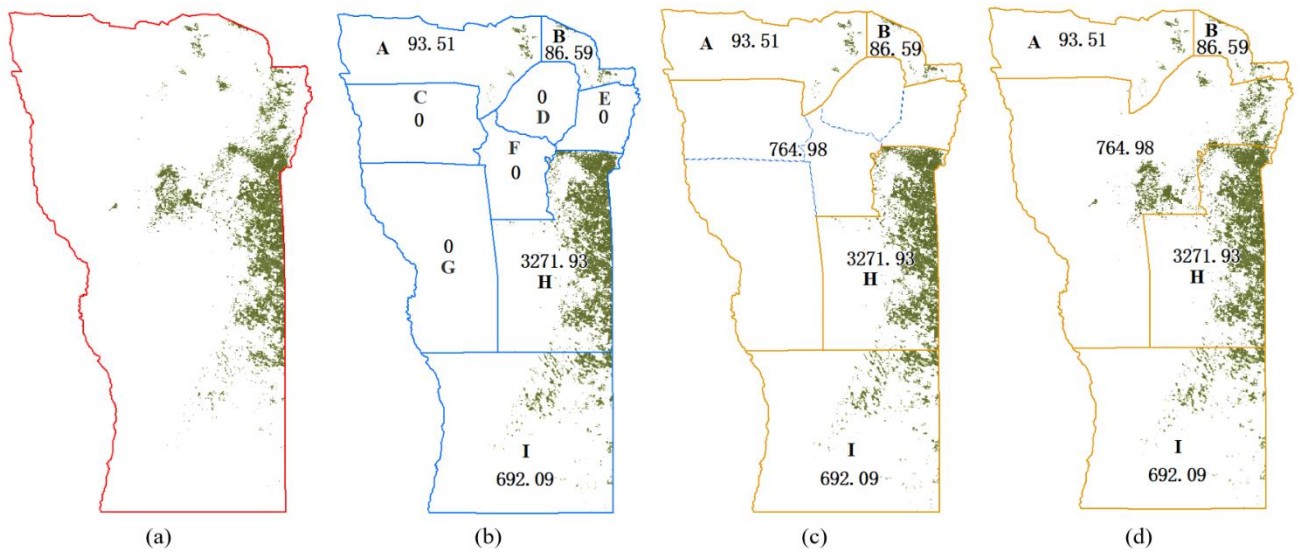

**Figure 3: The integration of the first and second subnational allocation results in San Luis Province, Argentina: (a) the first subnational allocation result, (b) the second subnational allocation result, (c) the combination of the departments with no statistics, and (d) the allocation results of the departments with no statistics.**

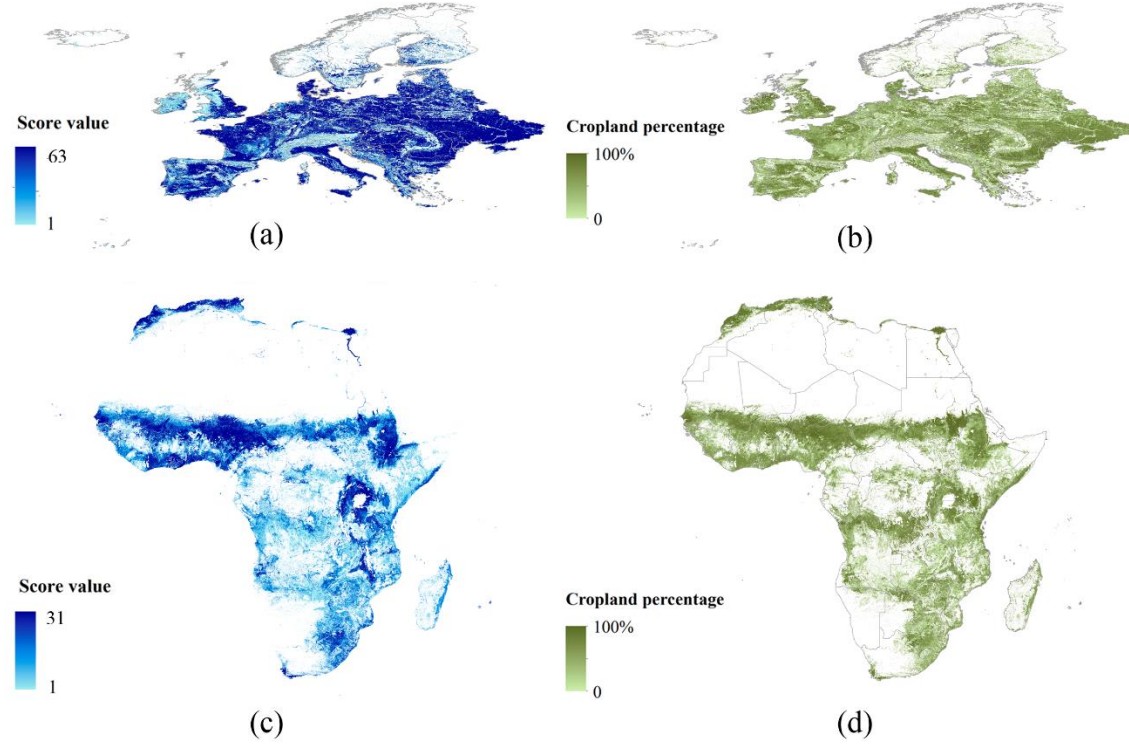


**Figure 4: Agreement ranking score maps and average cropland percentages in Europe and Africa: (a) and (b) are the score map and cropland percentage of Europe; (c) and (d) are the score map and cropland percentage of Africa.**

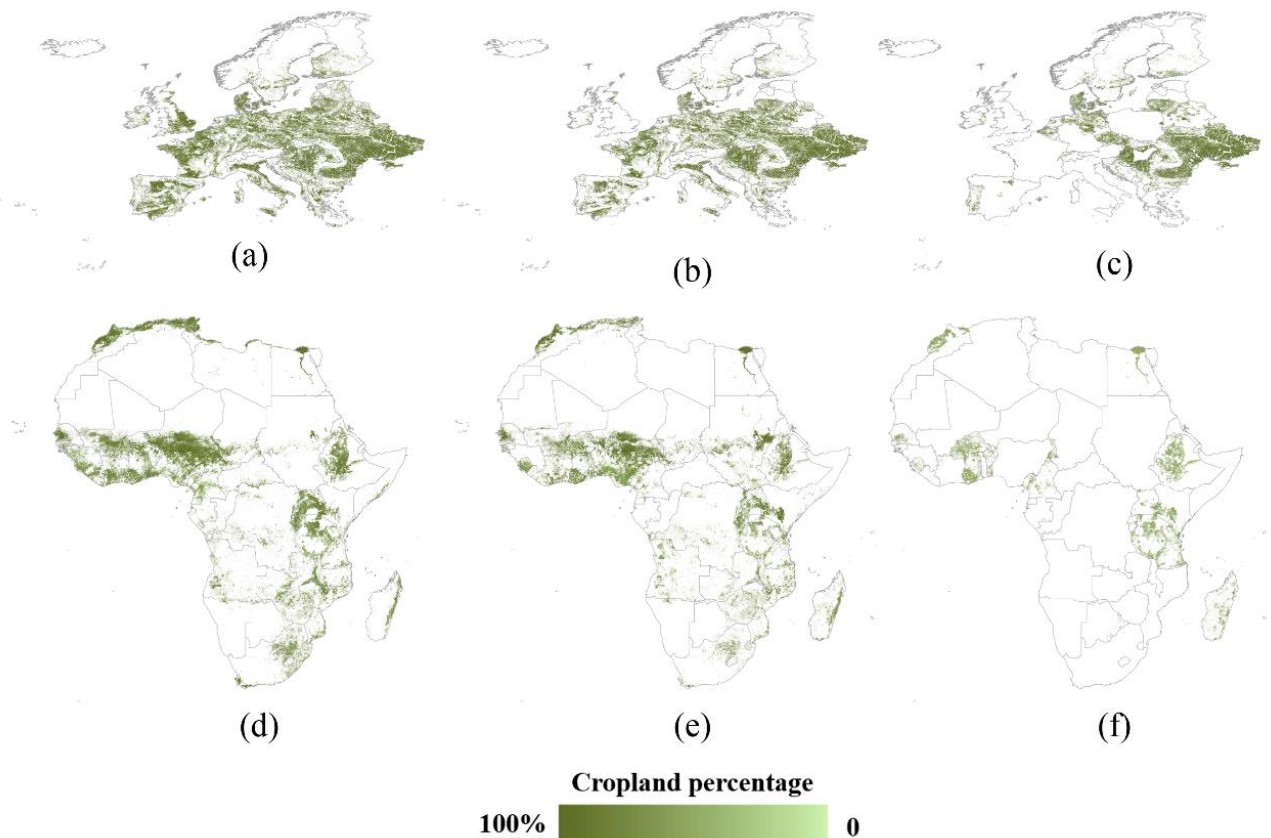

**Figure 5: Statistics allocation results in Europe and Africa: (a) and (d) are the national allocation results; (b) and (e) are the first subnational allocation results, (c) and (f) are the second subnational allocation results.**

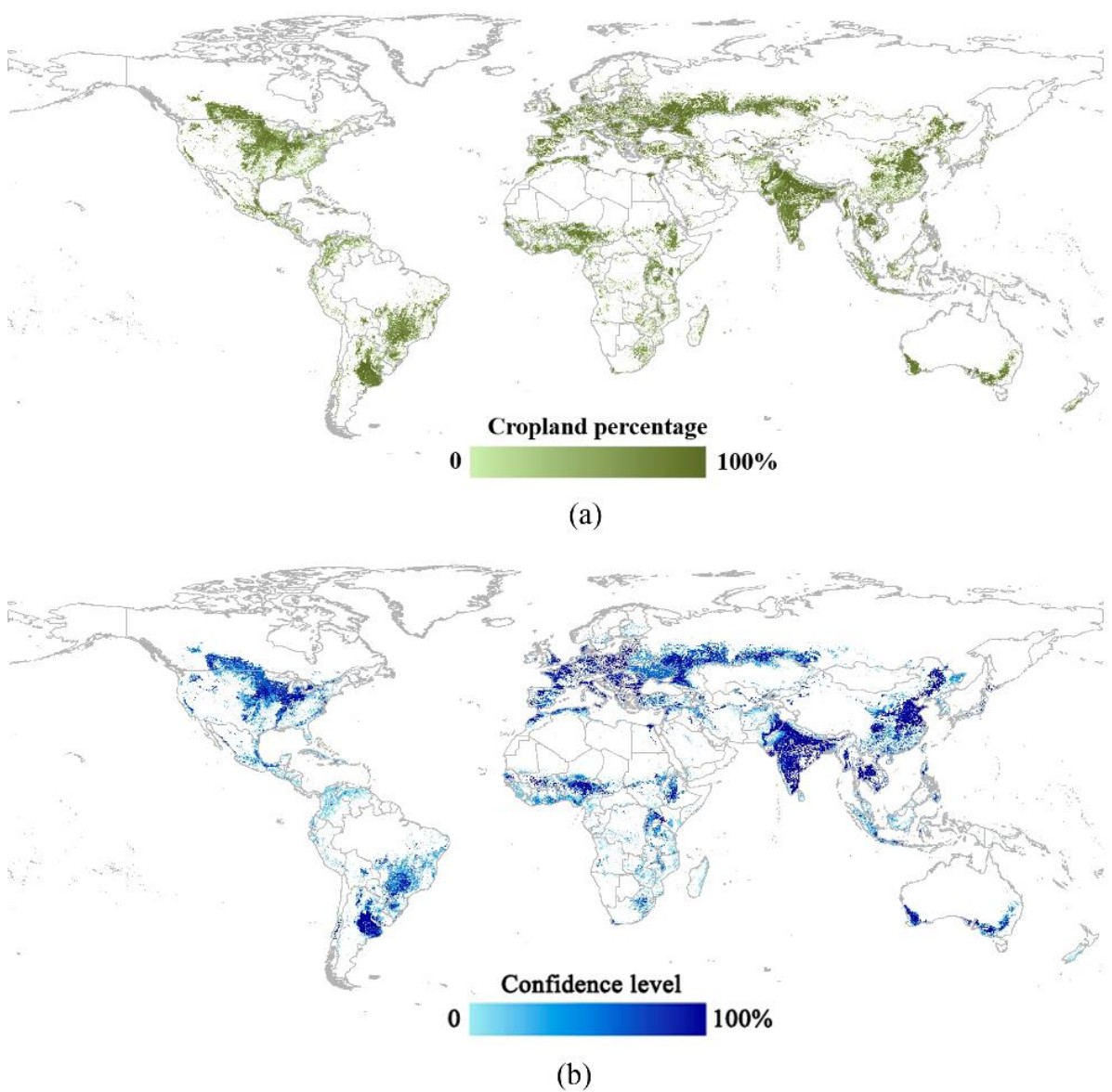

(a)

(b)

**Figure 6: The results of global synergy cropland: (a) cropland percentage map, (b) confidence level of synergy cropland.**


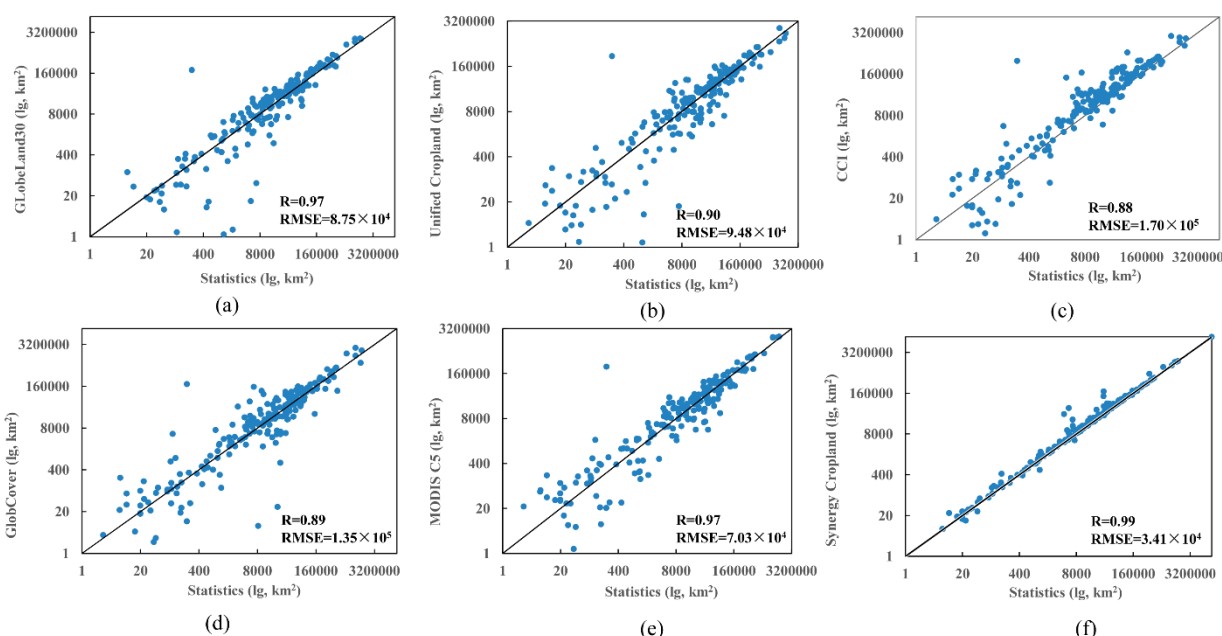

**Figure 7: The consistency analysis between cropland areas estimated from products and statistics: (a) GlobeLand30, (b) Unified Cropland, (c) CCI-LC, (d) GlobCover 2009, (e) MODIS C5, and (f) synergy map.**

**Table 1: Input satellite-based products.**

| Scale | Products | Time | Resolution | Producer & source |
|---|---|---|---|---|
| Global | GlobeLand30 | 2010 | 30 m | National Geomatics Center of China |
| | | | | http://www.globallandcover.com/GLC30Download/index.aspx |
| | CCI-LC | 2010 | 300 m | European Space Agency |
| | | | | http://maps.elie.ucl.ac.be/CCI/viewer/download.php |
| | GlobCover 2009 | 2009 | 300 m | European Space Agency |
| | | | | http://due.esrin.esa.int/page_globcover.php |
| | MODIS Collection 5 | 2010 | 500 m | Boston University |
| | | | | https://lpdaac.usgs.gov/products/mcd12q1v006/ |
| | Unified Cropland Layer | 2010 | 250 m | Université catholique de Louvain |
| | | | | https://figshare.com/articles/ucl_2014_v2_0_tif/2066742 |
| Regional | CORINE Land Cover (39 Europe countries) | 2012 | 100 m | European Space Agency |
| | | | | https://land.copernicus.eu/pan-european |
| | Land Cover of North America (Canada, USA, Mexico) | 2010 | 30 m | North American Land Change Monitoring System |
| | | | | http://cec.org/tools-and-resources/map-files/land-cover-2010-landsat-30m |
| Australia | The Land Use of Australia | 2010 | 50 m | Australian Government Department of Agriculture |
| | | | | http://www.agriculture.gov.au/abares/aclump |
| China | National Land Use/Cover Database | 2010 | 30 m | Chinese Academy of Sciences |
| | | | | http://www.resdc.cn/data.aspx?DATAID=99 |

**Table 2: The ranking scoring table for five input datasets.**

| Agreement level of input datasets | Score | A | B | C | D | E |
|---|---|---|---|---|---|---|
| 5 | 31 | 1 | 1 | 1 | 1 | 1 |
| 4 | 30 | 1 | 1 | 1 | 1 | 0 |
| | 29 | 1 | 1 | 1 | 0 | 1 |
| | 28 | 1 | 1 | 0 | 1 | 1 |
| | 27 | 1 | 0 | 1 | 1 | 1 |
| | 26 | 0 | 1 | 1 | 1 | 1 |
| 3 | 25 | 1 | 1 | 1 | 0 | 0 |
| | 24 | 1 | 1 | 0 | 1 | 0 |
| | 23 | 1 | 0 | 1 | 1 | 0 |
| | 22 | 0 | 1 | 1 | 1 | 0 |
| | 21 | 1 | 1 | 0 | 0 | 1 |
| | 20 | 1 | 0 | 1 | 0 | 1 |
| | 19 | 0 | 1 | 1 | 0 | 1 |
| | 18 | 1 | 0 | 0 | 1 | 1 |
| | 17 | 0 | 1 | 0 | 1 | 1 |
| | 16 | 0 | 0 | 1 | 1 | 1 |
| 2 | 15 | 1 | 1 | 0 | 0 | 0 |
| | 14 | 1 | 0 | 1 | 0 | 0 |
| | 13 | 1 | 0 | 0 | 1 | 0 |
| | 12 | 1 | 0 | 0 | 0 | 1 |
| | 11 | 0 | 1 | 1 | 0 | 0 |
| | 10 | 0 | 1 | 0 | 1 | 0 |
| | 9 | 0 | 1 | 0 | 0 | 1 |
| | 8 | 0 | 0 | 1 | 1 | 0 |
| | 7 | 0 | 0 | 1 | 0 | 1 |
| | 6 | 0 | 0 | 0 | 1 | 1 |
| 1 | 5 | 1 | 0 | 0 | 0 | 0 |
| | 4 | 0 | 1 | 0 | 0 | 0 |
| | 3 | 0 | 0 | 1 | 0 | 0 |
| | 2 | 0 | 0 | 0 | 1 | 0 |
| | 1 | 0 | 0 | 0 | 0 | 1 |
| 0 | 0 | 0 | 0 | 0 | 0 | 0 |


**Table 3: Cropland areas of each department from the first and second subnational allocation results, and their coordination in San Luis Province of Argentina.**

| Departments | Cropland area of the first subnational allocation result (km$^2$) | Cropland area of the second subnational allocation result (km$^2$) | Coordination of the two levels (km$^2$) |
|---|---|---|---|
| A | 93.51 | 93.51 | 93.51 |
| B | 86.59 | 86.59 | 86.59 |
| C | 1.87 | 0 | |
| D | 45.80 | 0 | |
| E | 496.55 | 0 | **4909.10–4144.12 =764.98** |
| F | 537.24 | 0 | |
| G | 84.15 | 0 | |
| H | 3271.93 | 3271.93 | 3271.93 |
| I | 291.46 | 692.09 | **692.09** |


**Table 4: Overall accuracies of input datasets and synergy cropland at the continent and global scales.**

|  | CCI-LC | GlobCover | GlobeLand30 | MODIS C5 | Unified Cropland | Synergy map |
|---|---|---|---|---|---|---|
|  | (%) | (%) | (%) | (%) | (%) | (%) |
| North America | 90.4 | 87.4 | 92.1 | 90.0 | 92.3 | 92.4 |
| South America | 78.8 | 78.9 | 90.1 | 87.5 | 89.7 | 89.4 |
| Europe | 89.7 | 87.5 | 87.1 | 89.4 | 88.6 | 93.7 |
| Africa | 79.1 | 83.1 | 89.9 | 88.7 | 86.1 | 89.1 |
| Oceania | 93.9 | 88.3 | 95.4 | 95.0 | 95.4 | 96.5 |
| Asia | 82.6 | 77.5 | 86.0 | 86.7 | 84.9 | 88.3 |
| Global | 84.5 | 83.0 | 89.3 | 88.8 | 88.1 | 90.8 |

**Table 5: Calculation of accumulated areas from high score value to low: (a) IIASA-IFPRI method, (b) SASAM.**

(a) IIASA-IFPRI method

| Score value | Accumulated area (km$^2$) | GlobeLand30 | Unified Cropland | GlobCover | CCI-LC | MODIS C5 | NLUC-C |
|---|---|---|---|---|---|---|---|
| 63 | $9.58 \times 10^5$ | 1 | 1 | 1 | 1 | 1 | 1 |
| 62 | $1.08 \times 10^6$ | 1 | 1 | 0 | 1 | 1 | 1 |
| 61 | $1.11 \times 10^6$ | 0 | 1 | 1 | 1 | 1 | 1 |
| 60 | $1.17 \times 10^6$ | 1 | 1 | 1 | 1 | 1 | 0 |
| 59 | $1.23 \times 10^6$ | 1 | 1 | 1 | 0 | 1 | 1 |

(b) SASAM

| Score value | Accumulated area (km$^2$) | GlobeLand30 | Unified Cropland | GlobCover | CCI-LC | MODIS C5 | NLUC-C |
|---|---|---|---|---|---|---|---|
| 63 | $9.58 \times 10^5$ | 1 | 1 | 1 | 1 | 1 | 1 |
| 62 | $1.08 \times 10^6$ | 1 | 1 | 0 | 1 | 1 | 1 |
| 61 | $1.14 \times 10^6$ | 1 | 1 | 1 | 0 | 1 | 1 |
| 60 | $1.17 \times 10^6$ | 1 | 0 | 1 | 1 | 1 | 1 |
| 59 | $1.19 \times 10^6$ | 0 | 1 | 1 | 1 | 1 | 1 |
| 58 | $1.25 \times 10^6$ | 1 | 1 | 1 | 1 | 1 | 0 |