# Peer review of "A cultivated planet in 2010: 1. the global synergy cropland map"

_Earth System Science Data, 2020_

## Referee Comment (RC1) · Anonymous Referee #1 · 11 Feb 2020

The paper by Lu and coauthors represents a very laudable effort to derive consensus maps on global cropland occurrence from multiple disagreeing input maps. This effort is highly valuable for the global scientific community. The importance of more accurate global cropland maps for research and applications in food security, climate modelling, and other fields cannot be overstated.

I see that this manuscript has already been in revision rounds before, and form what I can tell, the quality is already at a very high level at this point. I can confirm that the cropland maps are accessible and that the quality of the presented data is very high. I particularly like the spatial depiction of confidence levels. I was not sure, though, if these confidence maps are also being published alongside the cropland datasets (they should be!).

[Figure]

I have a couple of minor comments on how the manuscript and data and code presentation should be further improved:

I think the discussion could be slightly improved by including a section on necessary (or useful) next steps to further increase on the quality of the presented dataset.

I find the argument that the used statistical data on cropland are broadly reliable and therefore suitable as reference overly simplistic. While I do not imply to criticize the presented approach as such, I think it should be better acknowledged that also these statistics have uncertainties, and in some regions likely very large uncertainties. In some countries, individual remote-sensing based maps might even be more accurate than the used statistical data. Apart from the general acknowledgement of these uncertainties, could the authors maybe also add some discussion on where in the world the statistical data are deemed more or less reliable and why?

How exactly the confidence levels shown in Fig. 6 were generated is not fully clear to me. Could more information on this be provided (ideally with code to reproduce these maps or reapply the methodology elsewhere), including a short self-explanatory summary in the figure caption? Generally, I find it hard to interpret the figure captions without constantly referring back to the main text, so a few more explanatory words throughout the captions would be helpful.

The presented layers will surely be widely used, including in international policy contexts and as a reference for comparing or validating other products. Therefore, I think that absolute transparency regarding all input data and methodologies is required to assure reproducibility and possibilities for scientific scrutiny. In this respect, I have the following request:

To the extent that this is legally possible, all input census and point datasets that were integrated as a basis for developing the presented maps should be made openly accessible, either alongside the presented data products or via a separate publication. In cases where republishing these data may not be legally possible, sufficiently detailed

descriptions should be provided on how to independently access each input dataset so that the presented work could be scrutinized and reproduced by others. Similarly, sufficiently annotated code used to derive the presented layers from these source data, and code underpinning other presented analyses, should be provided.

I might have missed it, but I also didn't find the information on which data on cropping intensities were used to translate between harvested areas and crop areas (e.g., which values for which countries? What are the sources?) – these should be made fully transparent, too. Finally, although I ask for further transparency and some additional text, let me repeat that I regard this product as extremely valuable. My sincere gratitude to the authors for producing it and making it available!

---

## Referee Comment (RC2) · Anonymous Referee #2 · 21 Mar 2020

The authors created a global cropland map based on multiple LULC products by using the SASAM model. The synergy map presents high accuracy at global and regional levels, showing a promising use in the estimation of global agricultural production and policy-related managements, especially in the regions where lack of data.

I did not check the first draft of the paper, but I think the revised version has addressed the necessity of this work appropriately. All the sections are well presented. The different classification schemes incorporated in the multiple LULC datasets, as well as the uncertainties resulting from the pastureland, are also included in the discussion.

Therefore, I think the manuscript can be accepted for publication after the following minor questions are addressed

[Figure]

Minor revisions: Line 244, "... to consider than the national ...". Here I think it should be "... to consider that the national...". Line 249-269: The cropland distribution in Argentina in Fig.3. According to the rules, Fig.3C shows the merged cropland map based on the first and second subnational maps. My concern is since regions C-F have a total cropland area of 764.98 km2, what is the spatial distribution of these croplands? I am confused that I did not see the cropland distribution in regions C-F in Fig.3C. Maybe I did not quite understand the methods. Would you add some explanations about this?

Besides, it would be great if you can share your code of this work together with the synergy map.
* * *

---

## Author Comment (AC1) · 22 May 2020

Thank you for the comments concerning our Discussion Paper. These comments were very helpful for revising and improving our paper. We have responded to the comments point by point and made the detailed revisions embedded in the manuscript with the line numbers indicated in the responses.

Comment 1: I was not sure, though, if these confidence maps are also being published alongside the cropland datasets (they should be!).

Response: Indeed, the confidence map has been published with the cropland dataset, which is Figure 6(b) in the manuscript. Additionally, the confidence map is also posted on the website at: https://doi.org/10.7910/DVN/ZWSFAA.

[Figure]

Changes in manuscript: We added the website for the confidence map in Section 5 Data accessibility (line 420-421) of the manuscript.

Comment 2: I think the discussion could be slightly improved by including a section on necessary (or useful) next steps to further increase on the quality of the presented dataset.

Response: Thanks a lot for the constructive suggestions. We added our future works in the discussion. We will collect more input data and explore the integration of synergy approach and machine learning in the future to further improve the quality of the cropland dataset. The quantity and quality of input datasets are the basis of the synergy approach. We will collect more existing cropland maps with a high spatial resolution to refine the agreement ranking scores. SASAM accumulates cropland areas from high to low scores until the accumulated area reaches the statistics. The cumulative cropland area will be closer to the statistics with more input cropland datasets. Meanwhile, we will collect more statistics of cropland area at the subnational level. If all the subnational statistics at the global scale were available, the integration of multilevel allocation results would not be needed, which would greatly simplify the synergy process. To improve the method, we will explore the integration of synergy approach and machine learning according to the agreement of the input data and the geographical landscape. The synergy method is more economical and efficient for cropland mapping in those regions with relatively homogeneous landscapes. The regions with heterogeneous landscapes usually have lower agreements with higher uncertainties. Therefore, we will employ deep learning for cropland classification based on high spatial resolution images with the training samples from the agreements of existing cropland maps.

Changes in manuscript: Modification has been made to lines 481-494 of the manuscript.

Comment 3: Apart from the general acknowledgement of these uncertainties, could the authors maybe also add some discussion on where in the world the statistical data

are deemed more or less reliable and why?

Response: Thanks for the comment. Indeed, the quality of the cropland statistics varies. At the national level, the FAO statistical system (FAOSTAT) includes a quality framework and a mechanism to ensure the compliance of FAO statistics to this framework. Therefore, it is reasonable to assume that the national statistics have higher reliability. Subnational statistics at the global scale were collected from multiple sources, and the uncertainties are high because of various data processing and quality criteria. In Europe, America, Canada, and China the official censuses of cropland area at the subnational level are available and more reliable than those in Africa, Latin America and Asia, which have a weaker local capacity and the harvested areas of many crops are missing.

Changes in manuscript: We discussed the uncertainties of the statistics in the discussion (lines 475-480).

Comment 4: How exactly the confidence levels shown in Fig. 6 were generated is not fully clear to me. Could more information on this be provided (ideally with code to reproduce these maps or reapply the methodology elsewhere), including a short self-explanatory summary in the figure caption? Generally, I find it hard to interpret the figure captions without constantly referring back to the main text, so a few more explanatory words throughout the captions would be helpful.

Response: Thanks for these useful comments. We used the normalization of the score as the confidence level during the process of self-adapting statistics allocation. The value of the score indicates the agreement of the input cropland datasets, which reflects the confidence level of the cropland pixel. The scores range from 0 to 31 for five input datasets, and from 0 to 63 for six input datasets. Therefore, min-max normalization is used to normalize the scores to the same scale. The normalization results are the confidence levels with values from 0 to 100%.

Changes in manuscript: We added more explanation in Section 3.2 (lines 232 to 236),

and added the normalization step in the flowchart of cropland area statistics allocation with five input products (Fig. 2).

Comment 5: To the extent that this is legally possible, all input census and point datasets that were integrated as a basis for developing the presented maps should be made openly accessible, either alongside the presented data products or via a separate publication.

Response: The national statistics were acquired from FAO's FAOSTAT Land Use database (http://www.fao.org/faostat/en/#data/RL), which itself is open access. The subnational statistics were collected from various sources by IFPRI. We have made this dataset open access by providing it on this website: https://doi.org/10.7910/DVN/PRFF8V. The point datasets were provided to us by Tsinghua University. Unfortunately, we have a legal agreement with them that does not allow us to share it openly.

Changes in manuscript: We added the website for access to the subnational statistics to line 421-422.

Comment 6: Sufficiently annotated code used to derive the presented layers from these source data, and code underpinning other presented analyses, should be provided.

Response: All the code with annotations used for the synergy cropland mapping is shared at this website: https://sourceforge.net/projects/globalmapping/.

Changes in manuscript: We add the website of the code to the manuscript in lines 422-423.

Comment 7: I might have missed it, but I also didn't find the information on which data on cropping intensities were used to translate between harvested areas and crop areas (e.g., which values for which countries? What are the sources?) – these should be made fully transparent, too.

Response: This is an excellent question, and it has a complex answer. The original

subnational data are in harvested areas, and we converted harvested area into physical area (compatible with satellite-based cropland) by using cropping intensity. The cropping intensity varies by rainfed or irrigated systems and by countries. The data were collected from various sources such as seasonable harvested area, expert judgments and household surveys. In the companion paper "A cultivated planet in 2010: 2. the global gridded agricultural production maps" (Yu et al., 2020), the detailed description is in Section 6 of the supplementary information file.

Changes in manuscript: We added more description about cropping intensity to lines 165-167.

---

## Author Comment (AC2) · 22 May 2020

Thank you for the comments concerning our Discussion Paper. These comments were very helpful for revising and improving our paper. We have responded to the comments point by point and made the detailed revisions embedded in the manuscript with the line numbers indicated in the responses.

Comment 1: Line 244, "... to consider than the national ...". Here I think it should be "... to consider that the national...".

Response: Sorry for the writing error. We revised this error in line 253.

Changes in manuscript: We corrected the error in line 253.

[Figure]

Comment 2: Line 249-269: The cropland distribution in Argentina in Fig.3. According to the rules, Fig.3C shows the merged cropland map based on the first and second subnational maps. My concern is since regions C-F have a total cropland area of 764.98 km2, what is the spatial distribution of these croplands? I am confused that I did not see the cropland distribution in regions C-F in Fig.3C. Maybe I did not quite understand the methods. Would you add some explanations about this?

Response: Thanks for the comments. We added the spatial distribution for regions C-F with a cropland area of 764.98 km2 to Fig. 3(d). The self-adapting statistics allocation in Section 3.2 is a rerun for the merged departments of C, D, E, F, and G with a cropland area 764.98 km2. Based on the agreement ranking scores in Section 3.1, the cropland area 764.98 km2 is allocated to the pixels with higher ranking scores automatically until the cumulative cropland area is close to the 764.98 km2. Then, we obtained the allocation results of the merged region, as shown in Fig. 3(d).

Changes in manuscript: We added how to obtain the cropland distribution of regions C-F to lines 276 to 280, and the result is shown in Fig. 3(d).

Comment 3: Besides, it would be great if you can share your code of this work together with the synergy map.

Response: All the code with annotations used for the synergy cropland mapping is shared on this website: https://sourceforge.net/projects/globalmapping/.

Changes in manuscript: We added the website of the code to the manuscript in lines 422-423.